# An Ecological Validity Model for the Prevention of Obesity: Non-Nutritive Sweetener Consumption in Rats and the Effects of Switching from Sugar-Sweetened to Diet Beverages

**DOI:** 10.3390/nu14132758

**Published:** 2022-07-03

**Authors:** Heidi Morahan, Kieron Rooney

**Affiliations:** Faculty of Medicine and Health, Sydney School of Health Sciences, University of Sydney, Sydney 2006, Australia; heidi.morahan@sydney.edu.au

**Keywords:** non-nutritive sweetener, sugar-sweetened beverages, metabolism, obesity, behaviour, translational model

## Abstract

Reducing consumption of sugar-sweetened beverages (SSBs) has been encouraged due to its strong association with obesity. In parallel, consumption of “diet” or non-nutritive sweetened (NNS) beverages has significantly increased. This has led to burgeoning numbers of animal studies investigating metabolic consequences of NNS beverage consumption. However, most animal study designs do not reflect the way humans consume NNS drinks, thus reducing translational capacity. The present experiment aimed to find an ecologically valid model of NNS consumption and evidence of metabolic recovery following a switch from sucrose to NNS in female and male Sprague Dawley rats. The main behavioural outcome was consumption of commercially available NNS beverages during preference and acceptance testing, with changes to consumption following chronic sucrose consumption as a secondary outcome. The main metabolic outcome was retroperitoneal fat pad mass at culling, with body weight gain and fasting blood glucose levels (FBGLs) as secondary outcomes. In a two-phase experiment, behavioural tests were performed before and after 4 weeks of ad libitum access to 10% *w*/*v* sucrose. During Phase 2, the rats were given ad libitum access to assigned commercial NNS drinks for a further 4 weeks, with controls provided access to water only. FBGLs were measured at the end of Phases 1 and 2. Female and male rats accepted commercially available NNS beverages, although the volumes consumed varied considerably. Following the switch from sucrose to NNS (containing no sucrose), no group difference was observed in retroperitoneal fat mass, body weight change or FBGLs, suggesting both sexes exhibited limited metabolic recovery. These findings demonstrate that an ecologically valid model for NNS consumption can be developed for some commercially available NNS beverages to further enhance translational capacity.

## 1. Introduction

Global prevalence of obesity has dramatically increased over past decades and represents a major health challenge for both high- and low-income countries [1]. Worldwide, over 13% of the adult population are obese, and approximately 8% of deaths are the result of obesity as an attributing risk factor [2]. In 2016, it was estimated more than 1.9 billion adults and 340 million children and adolescents were considered overweight or obese, which is defined as abnormal or excessive fat accumulation to the degree that health may be impaired [3]. The World Health Organisation (WHO) and other international institutions classify an adult with a body mass index (BMI) between 25.0 and 29.9 kg/m^2^ as overweight, ≥30.0 kg/m^2^—as obese (for both sexes) [3,4]. Although the cause of obesity may be complex in nature, excessive consumption of added sugars has been widely recognised as a key contributor to its increasing prevalence [5,6]. Accordingly, global and public health authorities have strongly focused on reducing sugar-sweetened beverage (SSB) consumption [7,8] as these drinks represent the single largest source of added dietary sugars [9]. As such, non-nutritive sweetened (NNS) or “diet” beverages have been commonly advocated as a healthy alternative to SSBs. This is due to the fact they contain no or low calories, thus reducing energy intake while maintaining sweetness [10].

Despite the expected benefits of NNS consumption, evidence from some animal studies suggests NNSs may be associated with adverse metabolic effects, including obesity and disruption of glycaemic homeostasis [11,12,13]. In contrast, works of others have concluded NNSs show limited or no metabolic effects in mice and rats [14,15,16]. As the scientific debate continues, so too do animal studies. Rodents are frequently used in NNS research to further understand certain aspects of pathophysiology that may not be feasible in humans, such as invasive and end-of life investigations. In 2016, a large systematic review investigating low-energy sweetener (LES) consumption and body weight (BW) identified over sixty studies using either rat or mouse models [17]. Although this review did not formally assess study quality using a risk-of-bias tool, the authors concluded that most studies did not reflect the way humans consume LES beverages and questioned translational relevance. They found significant differences between the design of animal studies and human behaviours in their feeding patterns and in the volumes consumed. Supporting these findings, we also identified several design flaws in studies included in our recent systematic review investigating maternal NNS consumption in rodents. These included, but were not limited to, excessive NNS dosages (often exceeding daily intake limits for humans), inappropriate feeding models and poor sweetener choice [18]. We found that included studies were largely dominated by investigations of a single sweetener dissolved in water, commonly saccharin or aspartame. However, diet beverages consumed by humans generally contain two or more sweeteners blended to improve palatability. Moreover, saccharin makes up only a small portion of the NNSs in diet drinks consumed by humans [19], limiting the ability of such studies to assess relative risk to human health. We, therefore, considered how to improve study design investigating NNSs to better foster the translation of animal outcomes to human outcomes.

It is important future animal studies deliver enhanced translational capacity to ultimately reduce wastage of animals and resources [20,21]. Some encourage change to the way researchers design aspects of the animal diet that more adequately mimics human consumption, thus giving potential insights into human obesity [22]. We, therefore, wondered whether a more ecologically valid model of NNS consumption could be developed, that is, a consumption model in rodents that is more generalized to real-life settings. Consequently, this study aimed to test whether rats would drink commercially available NNS beverages commonly consumed by humans.

In addition to the above aim, the current study extended on previous work by our laboratory that tested the potential effect of NNSs to either prevent or mediate recovery from sugar damage [23]. That experiment used female Sprague Dawley rats and modelled a switch away from SSB consumption to an NNS solution. During the initial 4- or 8-week stage, all the rats were given unrestricted access to 10% *w*/*v* sucrose (representative of SSB concentration) in addition to water and chow. In the subsequent stage, two groups of rats undertook a dietary “switch” where sucrose was replaced with saccharin or water for a further 4 or 7 weeks. Metabolic measures for BW, insulin sensitivity and fat pad mass showed near complete recovery.

The present study design differs by providing both female and male rats with a dietary switch to commercially available beverages sweetened with NNSs, further mimicking human patterns of changing behaviour from drinking SSBs to diet beverages. Moreover, we added a test group that switched from sucrose to a reformulated blend of reduced sucrose and an NNS (Sprite™) to determine if a reduced sucrose/NNS mixture might also improve metabolic status. The primary metabolic measure to determine recovery was fat pad mass at culling.

It is common for NNS consumption to be tested in naïve animals, that is, rodents that have not been previously exposed to anything sweet [15,23]. Again, this does not appear to be reflective of how NNSs are consumed in the human population. Therefore, a secondary question was whether the consumption of commercially available drinks sweetened with NNS differed if animals had prior chronic exposure to sugar-sweetened drinks. The question we sought to answer was, would rats reduce or increase their consumption? This measure was assessed by performing preference and acceptance testing prior to and following sucrose feeding.

The experiment was conducted over two phases. In Phase 1, all the rats had access to a sucrose solution in addition to chow and water for a period of 4 weeks. We tested preference and acceptance of four different NNS beverages (and sucrose) before and after sucrose exposure. In Phase 2, the rats were allocated to the groups with four NNS beverages or water for a further 4-week period. In addition to fat mass, other metabolic measures included BW, BW gain and fasting blood glucose levels.

In summary, the primary aims of this study were twofold. First, to determine if rats consume commercially available NNS beverages, so an ecologically valid model for NNS consumption could be developed. Second, to determine if commercially available beverages sweetened with NNSs impact metabolic recovery from the damage caused by excess consumption of sugars.

## 2. Materials and Methods

### 2.1. Animals

Eighty experimentally naïve Sprague Dawley rats (40 females; 40 males) were block-randomised by weight and assigned using a computerised random number generator (RNG) to one of five groups (control, Diet Coke, Sprite, Cordial and Kombucha). Concealment of allocation to the groups was not possible due to the nature and delivery of dietary interventions. The animals were group housed (*n* = 4 rats/sex/cage) for a 7-day acclimatisation and handling period. They were maintained under controlled temperatures of 22–23 °C with humidity of 30–50% and standard 12:12 h reverse light/dark cycle (lights off at 10:00 h). The cages were randomly housed within the laboratory, and environmental enrichment was provided via cardboard tunnels and wood blocks for gnawing. Animals received ad libitum access to standard chow (Specialty Feeds^®^, Glen Forrest, WA, Australia; 14.2 kJ/g; 20% protein, 4% fat and 60% carbohydrate) and fresh tap water (Sydney water) throughout the study unless stated otherwise in the General Procedures. Approval was granted by the University of Sydney Animal Ethics Committee (No. 2019/1502). The protocol was registered in the preclinical trials registry (registry ID PCTE0000147; available from https://www.preclinicaltrials.eu/ Accessed on 23 March 2022).

### 2.2. Sample Size

A power analysis performed in G*Power (3.1.92, Heinrich Heine University Düsseldorf, Düsseldorf, Germany) was calculated using previous data from 24 h saccharin intake tests in naïve Sprague Dawley rats [23]. In addition, the effect size from fat pad mass data in Sprague Dawley rats following chronic sucrose feeding was also evaluated [23]. As such, a sample size of eight animals per treatment group was determined as optimal (α = 0.05; power of 95%).

### 2.3. Apparatus

Large racked acrylic individual ventilated cages (IVCs) were used for group housing (58 × 31 × 36 cm) with corncob bedding changed twice weekly. Smaller acrylic “drinking chambers” with metal wire lids were used for preference and acceptance testing (36 × 20 × 18 cm). Drinking solutions were accessible via plastic drinking bottles with metal spouts (500 mL for group cages and 100 mL for drinking chambers) inserted at designated openings on left and right sides. Chow was not available in the drinking chambers. Fluid intake was measured by weighing drink bottles on an electric balance to the nearest 0.1 g. Any visible spillages were recorded. Chow and fluid consumption per cage and individual bodyweight were measured every three days throughout the experiment.

### 2.4. Test Drinks

Four commercially available test drinks were chosen based on broad population consumption and common NNS combinations. The selected test drinks were Diet Coke™ (The Coca-Cola Company, Atlanta, GA, USA), Sprite™ (The Coca-Cola Company, Atlanta, GA, USA), Cottee’s Cordial No Added Sugar™ (Woolworths Ltd., Bella Vista, Australia) and Remedy Kombucha™ (Remedy Drinks, Melbourne, Australia). The beverages were de-carbonated using vigorous agitation on a magnetic plate stirrer, and Cordial was prepared as per manufacturer’s instructions of a 1-in-10 dilution. Solutions were prepared daily and offered at room temperature. Preference and acceptance experiments (Days 1–4 and Days 33–36) used solutions purchased on the same day to reduce potential batch discrepancies. The composition of beverages as described on the ingredients label is shown in Table 1. Additives were reported on labels as sweetener or food additive numerical codes. For the purpose of this table, numerical codes were converted to compound names according to the Food Additives and Sweetener Numerical List [24].

### 2.5. General Procedures

The experiment was conducted across two phases. Phase 1 was designed to assess preference and acceptance of test drinks and determine if a period of sucrose exposure will affect palatability and behavioural responses. Phase 2 switched experimental groups from sucrose to their assigned test drinks to assess the levels of potential metabolic recovery. The timeline of all the procedures is shown in Figure 1. BW was measured every three days during sucrose and NNS exposure, fluid intake—daily.

#### 2.5.1. Phase 1

Preference and acceptance testing of NNS drinks pre-chronic sucrose exposure: Prior to testing, the rats were acclimatised to individual drinking chambers and trained to drink from two bottles as previously described [25]. On Days 1 and 2, the animals undertook a 10 min preference test over two consecutive days (13:00 h) with water offered in one bottle and the assigned test drink in the other. The control group received 10% *w*/*v* sucrose. Water was placed on the left and the test drink was placed on the right for the first test session and vice versa for the second test session to account for potential position preferences. The bottle positions were counterbalanced across test sessions for all the groups. Following testing, the rats were returned to home cages and received 30 min water access. On Days 3 and 4, the rats undertook a 4 h acceptance test to measure the overall intake of the test drink. Acceptance was defined as an intake greater than 1 mL as this was also considered the error for spillage. Similarly to the preference testing, the animals were offered two bottles, one containing water and the other containing the test drink. Following behavioural testing, the animals received free access to water and chow.

Twenty-eight-day sucrose exposure (Days 5–33): Over the next four weeks, all the groups received ad libitum access to 10% *w*/*v* sucrose solution in addition to one bottle of water and chow. As previously reported [23], a four-week sucrose feeding model induces metabolic derangements in female rats, allowing identification of metabolic recovery or deterioration following a switch to the test drink during Phase 2. Due to large sucrose intake, bottles were measured and refilled daily. On the last day (Day 33), the rats were restricted to 30 min fluid access in preparation for preference testing which commenced the following day.

Preference and acceptance testing of NNS drinks post-chronic sucrose exposure: To observe if NNS consumption differed following chronic exposure to sugar-sweetened drinks, the preference and acceptance testing was repeated (Days 34–37).

Fasting glucose and insulin (Day 38): Following overnight fasting, the animals were randomly selected (using an RNG) for fasting blood glucose and plasma insulin and triglyceride testing by a blinded experimenter. Briefly, blood samples were taken from the lateral tail vein by removing the tail tip with a sterile scalpel. Blood glucose was determined with a hand-held glucometer (OneTouch Verio©, LifeScan IP Holdings, LLC, Malvern, PA, USA). An additional 100 µL of blood were collected and diluted 1:1 with 0.9% saline in an Eppendorf tube and centrifuged at 10,000 RPM at 4 °C for 20 min. Plasma was extracted and stored at −80 °C until further analysis of fasting insulin and triglyceride levels using a commercially available enzyme-linked immunosorbent assay (ELISA) kit (ALPCO Diagnostics, Salem, NH, USA).

#### 2.5.2. Phase 2

Twenty-eight-day NNS beverage exposure (Days 38–65): On the same day as blood sampling, the animals started 28 days of ad libitum drinking of one of the four test drinks (Diet Coke, Sprite, Cordial or Kombucha) or water (control) in addition to water and chow. Bottles were measured and refilled daily.

Fasting glucose, insulin and retroperitoneal fat mass (Day 66–68): Repeated blood sampling was taken using the exact methods as previously described (Section 2.5.1). The rats had free access to chow and water for 24 h prior to culling by lethal injection of 300 mg/mL sodium pentobarbitone (Valabarb^®^, Jurox, Rutherford, Australia). The animals were randomly divided (using a computerised RNG) between two cohorts, and retroperitoneal fat pads were excised and weighed by a blinded assessor over two consecutive days. Obesity was defined as a significant increase in fat pad mass relative to the control group, as currently there is no standard anthropometrical guideline defining rodent adiposity.

### 2.6. Data Analysis

All data are presented as the means ± SEM. Statistical analyses were performed using IBM SPSS Statistics version 28.0 (Armonk, NY, USA) with a significance level set at *p* < 0.05. Normality was assessed using the Shapiro–Wilk test for one-way ANOVAs and for violations of Mauchly’s test of sphericity in repeated and mixed measures; the Greenhouse–Geisser correction was used. The preference and acceptance tests were analysed using one-way ANOVAs with post-hoc pairwise comparisons applying Tukey’s HSD or the Games–Howell correction. To determine if chronic sucrose exposure changed an animal’s preference and acceptance to the test drinks, data before and after sucrose exposure were analysed using a 5 × (2) mixed ANOVA with Group as the between-subjects factor and Test as the within-subjects factor. During the 4-week sucrose exposure in Phase 1 when all the animals received identical treatment, repeated-measures ANOVAs were used to analyse changes in BW, chow and fluid consumption and total energy intake. Energy intake was calculated by converting chow (1 g = 3.4 kCal) and sucrose (1 g = 0.4 kCal) consumption into kilocalories. In Phase 2, when the groups were fed different test drinks, the data were analysed using 5 × (9) or 5 × (10) mixed ANOVAs with Group as the between-subjects factor and Days as the within-subjects factor. Changes in fasting glucose and insulin from the end of Phase 1 to the end of Phase 2 were analysed using 5 × (2) mixed ANOVAs with Group as the between-subjects factor and Test as the within-subjects factor. One-way ANOVAs with post-hoc comparisons were used to analyse FBGLs, FPI and retroperitoneal fat mass.

## 3. Results

### 3.1. Two-Bottle Preference Tests

Preference scores were calculated as the percentage of the consumed test drink relative to the total volume consumed (sum of the test drink and water) as a measure of preference for one beverage over water. An average was taken across the testing performed over two consecutive days, and data are shown for preference before and after sucrose exposure in Figure 2. Of note, both sexes displayed similar patterns of preference across most groups. As predicted, sucrose and Sprite were highly preferred over water, whereas the preference for Diet Coke was significantly lower, but still greater than that for Cordial and Kombucha, relative to water. Chronic sucrose exposure did not alter preferences for either sex.

#### 3.1.1. Test 1 Pre-Sucrose Exposure (Days 1–2)

Analysis of the preference scores found the main effect, indicating there was a significant difference for the test drink groups (female: F(4,35) = 94.10, *p* < 0.001; male: F(4,35) = 167.58, *p* < 0.001). Pairwise comparisons (Tukey’s HSD correction) revealed female preference for sucrose and Sprite was significantly higher than all other conditions (ps < 0.001), but did not differ from each other (*p* > 0.05), whereas in male rats, sucrose preference was greater than that for Sprite (*p* < 0.024). In both sexes, the preference scores for Diet Coke were greater than for Cordial and Kombucha (ps < 0.001), with no difference detected in the latter two conditions in either sex, F > 1.

#### 3.1.2. Test 2 Post-Sucrose Exposure (Days 34–35)

Mean scores provided in repeated preference tests following 28-days of sucrose exposure were analysed using 5 × (2) mixed ANOVAs. In females and males, there was no significant Group × Test interaction detected (ps > 0.05), indicating preference from Test 1 to Test 2 did not differ between groups (see Figure 2).

### 3.2. Four-Hour Acceptance Testshis 

Intake of the test drinks over a 4 h period before and after sucrose exposure is shown in Figure 3. As expected, sucrose and Sprite intakes were significantly higher than that of other test groups for both sexes. An interesting observation from the data was that, although naïve female and male rats drank similar volumes of sucrose and Sprite during their tests, when adjusted for BW, females drank significantly more than males, whereas the body weight-adjusted intakes for the test drinks that did not contain sucrose (Diet Coke, Cordial and Kombucha) did not differ between sexes.

#### 3.2.1. Test 1 Pre-Sucrose Exposure (Days 3–4)

One-way ANOVAs revealed a group difference in test drink intake (females: F(4,35) = 61.27, *p* < 0.001; males: (F(4,34) = 25.58, *p* < 0.001). One dataset from the male sucrose group was excluded from analyses due to a drink bottle blockage. Pairwise comparisons confirmed the intakes of sucrose and Sprite were significantly higher than those in the other groups (ps < 0.004), although they did not differ from each other (ps > 0.05). This was evident for both sexes. In females, consumption of Diet Coke, Cordial and Kombucha did not differ over a 4 h period (ps > 0.05), whereas males drank more Diet Coke than Cordial (*p* < 0.005) (see Figure 3a). When the intakes were adjusted for BW (intake g/kg BW), a 5 × (2) (test drink × sex) ANOVA identified the main effect of sex, where female rats drank more sucrose (F(1,68) = 39.00, *p* < 0.001) and Sprite (F(1,69) = 37.20, *p* < 0.001) than male rats. There was no sex effect on weight-adjusted intakes in the Diet Coke, Cordial or Kombucha groups, (ps > 0.39) (see Figure 3b).

#### 3.2.2. Test 2 Post-Sucrose Exposure (Days 36–37)

Following 4 weeks of sucrose exposure, analysis using 5 × (2) mixed ANOVAs observed a Group × Test interaction (females: F(4,35) = 2.96, *p* < 0.033; males: F(4,34) = 6.78, *p* < 0.001). Post-hoc comparisons revealed a simple main effect for Group, indicating rats drank more sucrose (females: F(1,7) = 10.75, *p* = 0.014; males: F(1,6) = 9.61, *p* = 0.021) and Sprite (females: F(1,7) = 8.23, *p* = 0.024; males: F(1,7) = 27.67, *p* = 0.001) during Test 2 compared to Test 1 (See Figure 3a). No difference was detected for the other conditions (ps > 0.05).

### 3.3. Body Weight and Weight Gain

BW data are presented in Figure 4. During Phase 1, mean BW increased significantly from 184.4 ± 2.3 g to 260.0 ± 4.7 g in females (linear trend: F(1,35) = 759.90, *p* < 0.001) and from 305.9 ± 2.9 g to 476.9 ± 7.3 g in males (linear trend: F(1,35) = 918.19, *p* < 0.001). One-way ANOVAs found no difference in BW or weight gain on the final day of Phase 1 for either sex (Fs < 1). At the end of Phase 1, the animals were fluid- and chow-restricted in preparation for behavioural and physiological testing. This was reflected in a small BW reduction on the first day of NNS exposure (NNS1), although this soon recovered. Similarly, BW increased linearly during Phase 2 (females: F(1,35) = 554.00, *p* < 0.001; males: F(1,35) = 2131.87, *p* < 0.001). However, analyses failed to detect a Day × Group interaction (ps > 0.26), indicating that the rate of weight gain did not differ between the test drink groups. Baseline BWs during test group allocation are displayed in Table 2 and Table 3.

### 3.4. Consumption and Energy Intake

Because rats were group-housed, consumption data were obtained by dividing the overall cage fluid or chow intake by the number of animals per cage (*n* = 4) and reporting as mean g/rat/day or mL/rat/day. The power to detect group differences in consumption was limited as there were only two cages/sex/test group. Thus, some statistical analyses were not reported, and the mean group intakes are presented in Table 2 (for females) and Table 3 (for males).

#### 3.4.1. Phase 1: Sucrose and Chow Consumption, Energy Intake

As expected, all the rats drank almost exclusively the sucrose solution during 28 days of exposure (mean water intake = 0.8 mL/day). In both sexes, sucrose intake steadily increased over days (female linear trend: F(1,35) = 141.39, *p* < 0.001; male linear trend: F(1,35) = 191.86, *p* < 001). In contrast, chow intake decreased over the same period (female linear trend: F(1,35) = 880.75, *p* < 0.001; male linear trend: F(1,35) = 343.32, *p* < 0.001), indicating a compensatory effect for the increased energy contribution from sucrose by reducing chow consumption. During Phase 1, the average daily energy intake was 72.3 kCal/rat/day for females, with sucrose contributing an average of 31.8 kCal/rat/day (44.0% of the total daily energy intake), whereas the daily energy intake for males was significantly greater (*p* < 0.05), with an average of 111.3 kCal/rat/day and sucrose contributing 37.2 kCal/rat/day (33.4% of the total daily energy intake).

#### 3.4.2. Phase 2: NNS Drink and Chow Consumption, Energy Intake

Following the switch from sucrose to NNS, an interesting observation was that the total average daily fluid intake (water plus test drink) for most NNS test drink groups remained higher than in the control group where only water was offered, suggesting a pattern of over-drinking. This was particularly evident in the Sprite group (females: 53.5 ± 1.7 mL/rat/day; males: 81.1 ± 1.3 mL/rat/day) compared to the control group (females: 21.4 ± 0.4 mL/rat/day; males: 33.5 ± 0.4 mL/rat/day) (see Figure 5a). The percentages of test drink contribution to the average daily fluid intake recorded for females and males, respectively, for Diet Coke were 62.5% and 65.8%, 95.1% and 95.6% for Sprite, 26.5% and 27.2% for Cordial, 32.0% and 28.3% for Kombucha.

The mean chow consumption during Phase 2 is shown in Table 2 and Table 3. Across measurement periods in Phase 2, 5 × (9) ANOVA analyses were performed. In females, this failed to detect any Group × Day interaction during NNS exposure (F < 1). However, a significant main effect for Group was revealed, with pairwise comparisons indicating the Sprite group ate significantly less than other groups (*p* = 0.012) and the Kombucha group consumed more chow than the control and Sprite groups (ps < 0.036). In males, an interaction between Day and Group (F(4,35) = 6.44, *p* < 0.001) and the main effect of Group (F(4,35) = 14.93, *p* < 0.001) were detected. Further analysis found the Sprite group ate less than all the other groups (ps < 0.01). We also compared the average daily chow intake between Phases 1 and 2 using separate 5 × (2) ANOVAs. A significant interaction between Group and Phase was detected (females: F(1,39) = 15.92, *p* < 0.005; males: F(1,39) = 8.52, *p* < 0.001), indicating a difference in chow intake between the groups from Phase 1 to Phase 2. Post-hoc analyses revealed chow consumption increased in all the groups (female *p* < 0.046; male *p* < 0.012), except for the Sprite group where consumption remained unchanged (See Figure 5b). Similar analyses were conducted to determine if the total daily energy intake differed between Phases 1 and 2. A significant reduction in energy intake was seen in all the groups, with the exception of the male Sprite group which did not change (*p* > 0.05).

### 3.5. Fasting Blood Glucose and Plasma Insulin

Fasting blood glucose levels are shown in Figure 6. At the end of Phase 1, when all the animals were treated in an identical way, one-way ANOVA analyses confirmed there was no significant difference in FBGLs between the groups (females: *p* = 0.24; males: *p* = 0.13). Subsequently, when the rats were switched to the allocated test drinks for a further 28 days (Phase 2), 5 × (2) mixed ANOVA analyses found Group × Phase interactions (females: F(4,35) = 4.71, *p* = 0.04; males: F(4,35) = 3.31, *p* = 0.021). In females, further analyses revealed a simple main effect for Group (F(4,35) = 3.99, *p* < 0.009) and Phase (F(1,7) = 11.46, *p* = 0.012), with the pairwise comparison identifying the Sprite group had both significantly higher FBGLs than all the other groups in Phase 2 (ps = 0.04) and increased between the tests (*p* = 0.012). No difference was detected between the latter conditions. Similarly, a simple main effect for Phase was identified in males, with post-hoc analyses finding control (F(1,7) = 14.61, *p* = 0.007) and Sprite (F(1,37) = 6.91, *p* = 0.034) FGBL values increased between the tests. No other interactions or effects were detected.

Due to technical issues, female plasma insulin samples could not be analysed. Mean plasma insulin data for males are displayed in Table 3. Two plasma samples, one Sprite and one Diet Coke, were haemolysed and could not be analysed; therefore, these male groups had *n* = 7 for insulin analyses. Analysis of change from Phase 1 to Phase 2 detected no significant Phase × Group interaction or main effect of Phase (ps > 0.10). A one-way ANOVA conducted on male insulin data at the end of Phase 2 identified the main effect of Group (F(4,33) = 4.66, *p* = 0.004). Post-hoc analysis revealed the Sprite group had significantly higher insulin values than the control group (*p* = 0.002) but did not differ from the Diet Coke or Cordial groups (ps > 0.17). A trend was noted for the Sprite group insulin data to be greater than in the Kombucha group (*p* = 0.052).

### 3.6. Retroperitoneal Fat Pad Mass

Retroperitoneal fat mass was adjusted for BW (g/kg) and is shown in Table 2 and Table 3. One-way ANOVAs were conducted separately for female and male cohorts. No significant difference of fat mass at culling was detected: females: F(4,35) = 1.14, *p* = 0.36; males: F(4,35) = 1.12, *p* = 0.36.

## 4. Discussion

The main aim of this experiment was, first, to determine if rats would drink commercially available NNS beverages commonly consumed by humans, thus to develop an ecologically valid animal model for future obesity research. In terms of acceptance of these “diet” drinks, the results were largely positive, with both female and male rats drinking the NNS beverages offered. However, the volume consumed and preference for different test beverages varied considerably. As for the second aim, we assessed whether switching from SSBs to NNS beverages could prevent or mediate metabolic recovery from sugar damage. Earlier assessments conducted in female rats switched from sucrose to saccharin suggest almost complete recovery is possible [23]. However subsequent replication studies failed to confirm this [26]. Given the use of rarely consumed NNSs in the diet of humans limits translational capacity of that work [19], we undertook a different study design to better mimic human NNS consumption patterns. Below, we discuss our findings for the related outcome measures in more detail.

Developing an ecological validity model of NNS consumption in rats is dependent on whether animals accept and prefer the offered beverages. We had predicted, and our results confirmed, that animals would consume Sprite on the basis that it contains sucrose and stevia, both highly preferred by rodents [27] and containing a higher energy content for greater satiety. Some studies have reported on metabolic effects of Diet Coke consumption in rodents; however, behavioural testing and detailed beverage intake assessments were not performed [28,29]. In regard to the present study, our preference and acceptance results established Sprite was highly preferred and accepted, followed by Diet Coke, whereas both Kombucha and Cordial had very low intakes. Of note, considerable individual differences were seen in some animals, mainly in the Cordial, Diet Coke and Kombucha groups. Such variability was expected as differences in behavioural responses to flavoured solutions have been well-reported [30,31]. Despite this, the average fluid intakes during Phase 2 remained consistent with the drinking patterns displayed in the behavioural testing. That is, the rats consumed more Sprite than Diet Coke and more diet coke than Kombucha and Cordial. Another interesting observation was the trend of over-drinking when the rats were offered both water and NNS beverages; however, this was not as evident in the beverages that did not contain sucrose. Caution should be taken when drawing conclusions on this observation as it may be specific to rodents and we cannot generalise that this effect is representative of human drinking behaviours.

In line with enhancing translational outcomes, the present study design also explored whether consumption of commercially available drinks sweetened with NNSs differed if animals have had prior chronic exposure to sugar-sweetened drinks. Repeated preference and acceptance test post-sucrose feeding failed to confirm this effect. Only male rats in the Sprite group and the control group (given sucrose in the behavioural testing) increased intakes during acceptance testing after four weeks of sucrose exposure. In summary, while each of the commercially available NNS beverages was accepted by the rats (previously defined as intakes greater than 1 mL), we determined that sugar-free Cordial and Kombucha were not reliable ecologically valid models for NNS consumption due to low intake volumes.

With regard to metabolic outcomes after switching from sucrose to NNS beverages, our primary outcome measure was a change in body fat, specifically retroperitoneal fat pad mass. Separate studies have reported conflicting results. Previous work by our laboratory demonstrated female rats showed an improved metabolic profile, including retroperitoneal and visceral fat pads, following a switch from chronic sucrose feeding to saccharin or water [23]. However, in more recent experiments where sucrose-fed rats were switched to water alone, both sexes showed persistence of adiposity, concluding only a partial recovery was made [26]. It was not unexpected for males to display persisting body fat given previous observations [32], but our current experiment supports the conclusion that females also showed incomplete recovery where adiposity remained similar between the control and test drink groups, as seen in Table 2 and Table 3. This is consistent with measures of body weight and weight gain, where no detectable change between the groups was noted at the end of Phase 2. We must also take into account that, although rodent fat pad mass is commonly used as a measure for human adiposity, there may be limitations when comparing fat distribution and function between species [33].

As for other secondary metabolic measures, blood marker results demonstrate only the rats that continued to drink sucrose for a further 4 weeks, albeit at a reduced concentration (i.e., the Sprite group), showed an increase in fasting blood glucose levels in both sexes. For females, this value was higher than in all the other groups, suggesting the detrimental effect of sucrose was even greater. Unfortunately, female plasma insulin samples could not confirm this. Gender differences in blood glucose and plasma insulin levels have been reported, with females more susceptible to poorer outcomes [34]. Male insulin data were inconsistent, with the Sprite group higher than controls but not differing from other groups that switched to NNS beverages. This suggests the model of changing from sucrose to diet drinks does not mediate recovery in males, although there is the possibility we did not have adequate power to conclude this result. A secondary consideration is whether recovery in rats was not achieved due to NNS adversely effecting metabolism; however, the possibility is reduced due to low intake volumes in the Kombucha and Cordial groups. Some studies have reported changes to the gut microbiome following NNS consumption, which may be associated with poor glycaemic outcomes [12,35], whereas others have contradicted this [36]. We cannot confirm or refute this result as it was outside the scope of our research aims. Moreover, a limitation of this current study is the lack of inclusion of a water/chow group, so determination of sucrose-induced metabolic damage during Phase 1 could not be presented.

An important feature of this study is the modelling of human consumption to enhance translational outcomes, which has historically been poor [37,38]. As such, we attempted to develop an improved ecologically valid model to inform further studies and prevent an unnecessary use of animals under the guise of informing human health. Commercially available diet beverages contain other ingredients, as listed in Table 1, and we cannot be certain that these components do or do not have physiological interactions that affect metabolism. However, it is important to recognise that a translational consumption model in animals should mimic human consumption.

Despite our attempts to enhance translation through an ecologically relevant experimental design, another important consideration is the question of predictability of animal-to-human outcomes, specifically in the context of obesity. Of course, we cannot be certain of translational success of our results; however, prior to developing the study design, factors were identified to address external validity. The selection of an appropriate animal model to simulate human disease is a critical step in animal studies [39]. Our meta-analysis in maternal NNS consumption provided quantitative data showing rat models displayed effects in body weight changes compared to mouse models, although, on balance, much of these data were predisposed to bias [18]. Moreover, Sprague Dawley rats are commonly used in obesity-related research as they have been shown to display diet-induced adiposity similar to humans [40]. Concurring with this, prior work conducted in our laboratory which provided female and male Sprague Dawley rats ad libitum 10% sucrose solution in addition to chow and water showed metabolic changes similar to those produced in humans by a poor diet [32]. Taken together, best efforts were made to select an appropriate animal model so the results from this study would be more relevant when considering strategies to prevent human obesity, such as replacing SSBs with “diet” beverages. Although we identified limited metabolic recovery following the switch, we cannot be certain full recovery could not be achieved given a longer period of NNS consumption. Of considerable interest for future investigations is the recent trend whereby manufacturers have reduced sugar content in beverages, as was the case with Sprite™. Our findings suggest chronic sucrose consumption at reduced concentrations does not attenuate the deleterious effects of sugars. However, our study did not explore the potential metabolic differences between 10% sucrose and a reduced sucrose model, which may be a direction for future research.

In conclusion, this experiment shows that an ecologically valid model for the consumption of NNSs can be developed, with female and male rats consuming Diet Coke and Sprite in larger volumes than sugar-free Cordial and Kombucha. In addition, we found both sexes displayed limited metabolic recovery after switching from sucrose to “diet” beverages.

## Figures and Tables

**Figure 1 nutrients-14-02758-f001:**
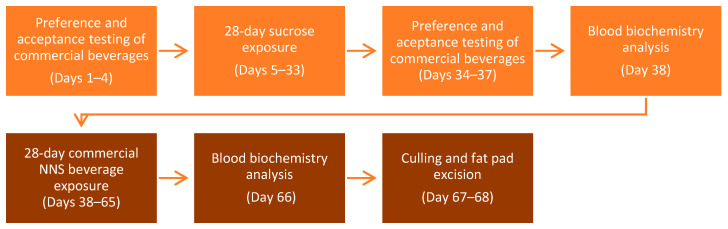
General timeline of all the procedures during the two phases of the experiment.

**Figure 2 nutrients-14-02758-f002:**
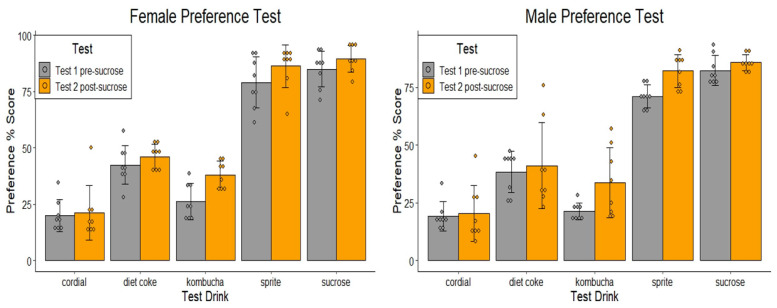
Preference tests performed in female and male naïve animals (Test 1 pre-sucrose) and following 4 weeks of sucrose exposure (Test 2 post-sucrose) during Phase 1. Mean % (± SEM) preference of the assigned test drink over water during 10 min two-bottle preference tests. The control group received 10% *w*/*v* sucrose (*n* = 8/group).

**Figure 3 nutrients-14-02758-f003:**
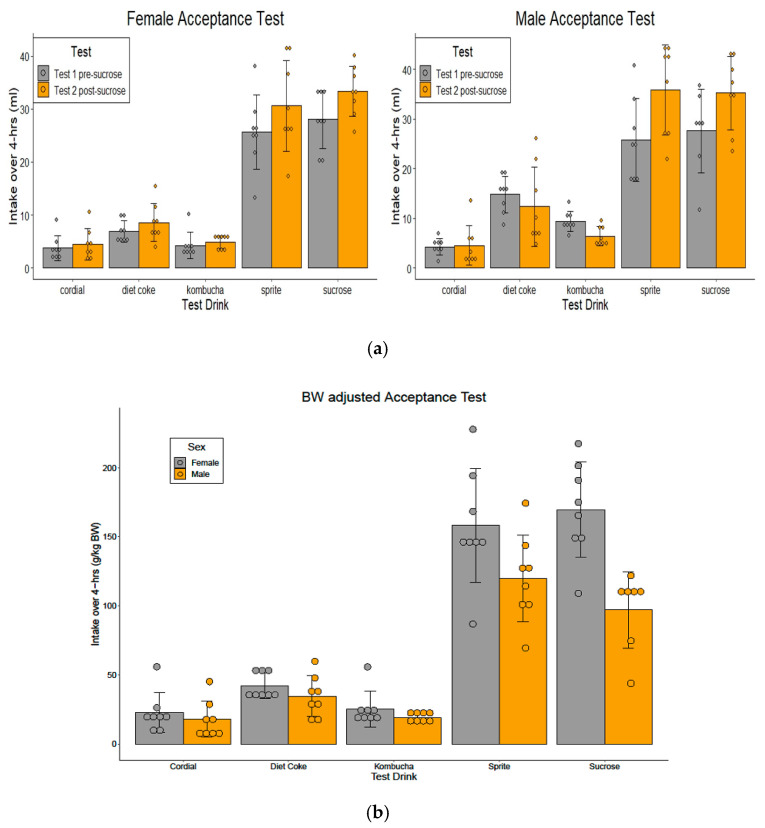
(**a**) Acceptance tests performed in female and male naïve animals (Test 1 pre-sucrose) and following 4 weeks of sucrose exposure (Test 2 post-sucrose) during Phase 1. Mean intake (mL ± SEM) of the test drinks during a 4 h acceptance test showed female and male rats drank significantly higher amounts of sucrose and Sprite compared to Diet Coke, Cordial and Kombucha, *p* < 0.001, which had similar intakes. Sucrose and Sprite intakes increased following 4 weeks of sucrose exposure (*p* < 0.05) in both sexes. (**b**) BW-adjusted acceptance test in female and male naïve animals during Phase 1. Mean intake (g/kg BW) was significantly higher in female rats in the sucrose and Sprite groups (*p* < 0.05) (*n* = 8/sex/group; *n* = 7 males in the sucrose group).

**Figure 4 nutrients-14-02758-f004:**
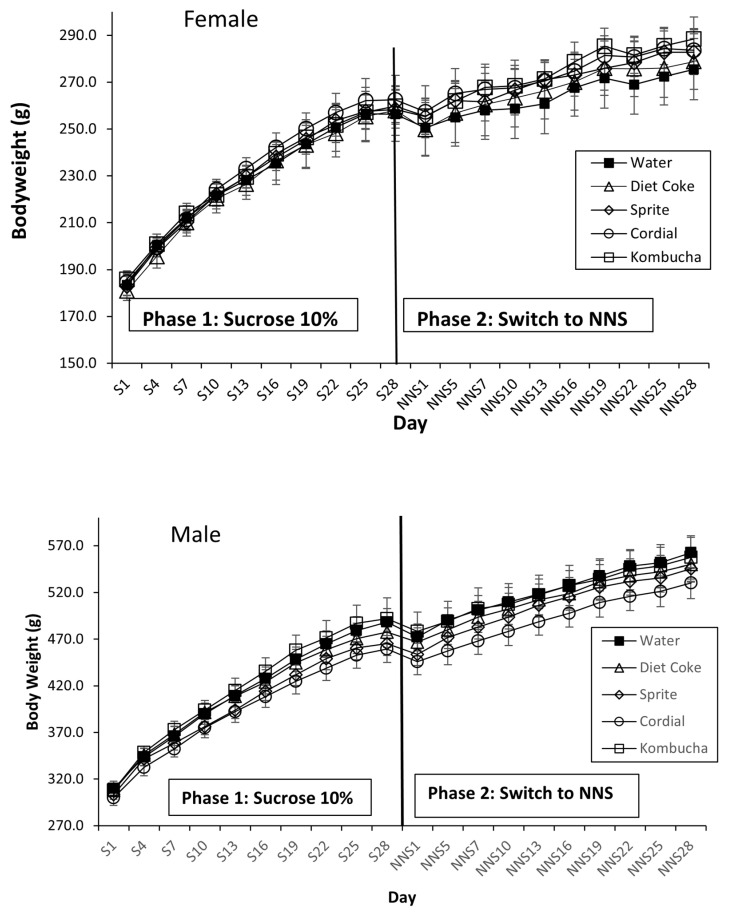
BW gain for the groups during Phase 1 sucrose exposure and following a switch to NNS exposure in Phase 2 in female and male rats. During Phase 1, all the rats received 28 days of 10% sucrose in addition to water and chow (*n* = 40). In Phase 2, the rats received 28 days of their assigned NNS test drink in addition to water and chow (*n* = 8/group). No significant effect on BW was observed at any point (*p* > 0.05). S = sucrose; NNS = non-nutritive sweetener.

**Figure 5 nutrients-14-02758-f005:**
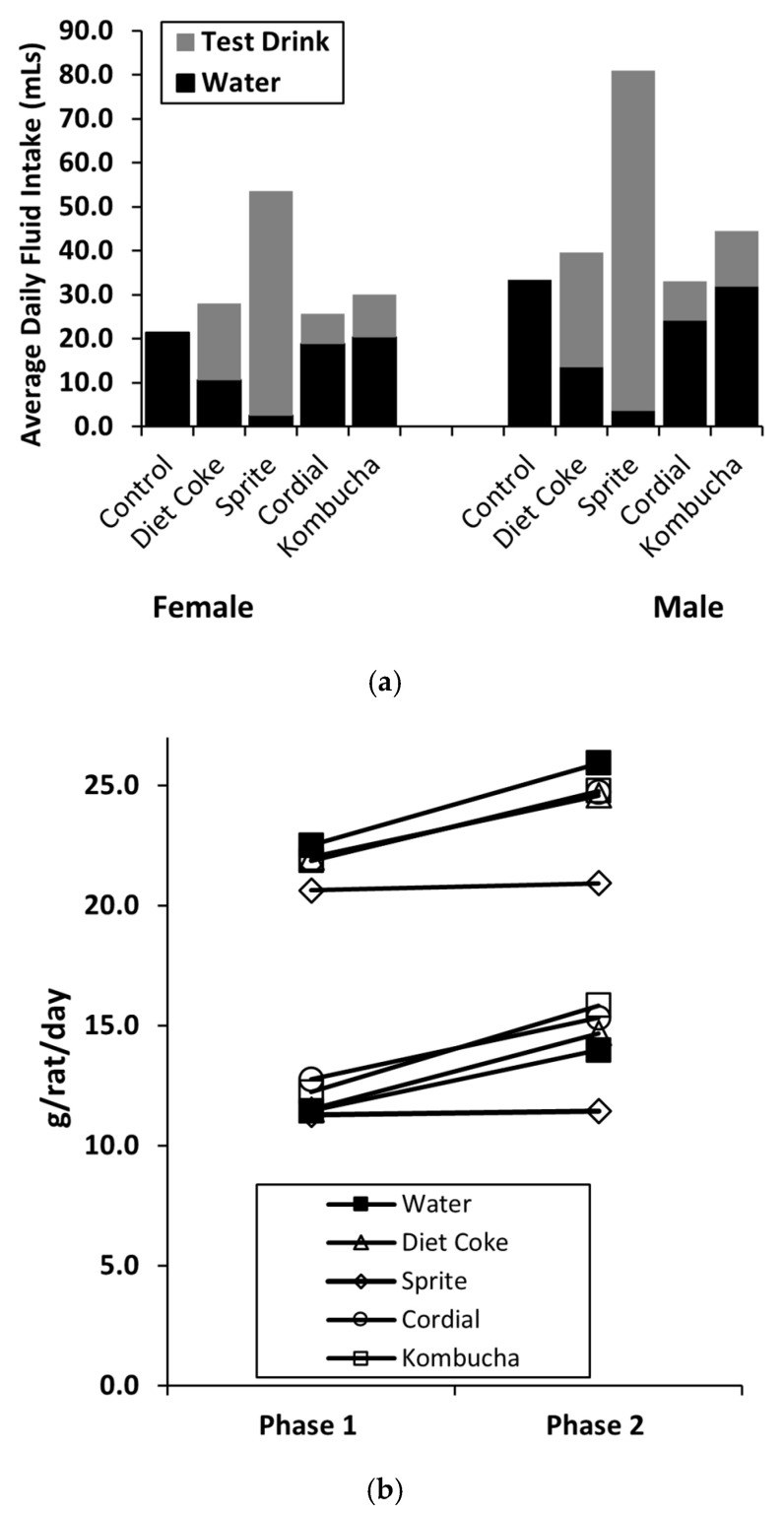
Phase 2 consumption data for the mean total daily fluid intake in Phase 2 (**a**) and chow intake (**b**). Sprite was highly consumed during 28 days of NNS exposure, followed by Diet Coke, Kombucha, and Cordial. Chow consumption increased for all the groups except Sprite, where it remained unchanged (*n* = 8/group).

**Figure 6 nutrients-14-02758-f006:**
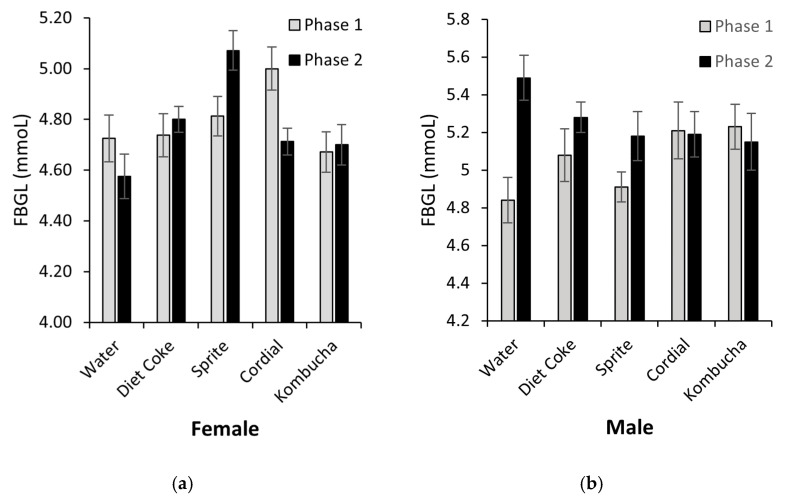
Fasting blood glucose levels measured at the end of Phases 1 and 2 for female (**a**) and male rats (**b**). Female Sprite group’s FBGLs increased between the tests (*p* = 0.012) and were significantly higher than in the other groups (*p* = 0.04). Male Sprite and control groups’ FBGLs increased between the tests (*p* < 0.34) (*n* = 8/sex/group).

**Table 1 nutrients-14-02758-t001:** Components and nutritional information of the commercially available drinks administered during preference and acceptance testing and the four-week period following chronic sucrose feeding.

	Diet Coke™	Sprite™	Cottee’s Cordial Raspberry (No Added Sugar)™	Remedy Kombucha Ginger and Lemon™
Sweetener(s) *	aspartame acesulfame K	sucrose stevia	cyclamate acesulfame K sucralose	stevia erythritol
Other ingredients *	water, colour (caramel IV), caffeine, phosphoric acid, citric acid	water, citric acid, sodium citrate, flavour, potassium sorbate	water, citric acid, flavour, sodium carbomethylcellulose, sodium benzoate, sodium metabisulphites, colour (carmoisine)	water, kombucha culture, black tea, green tea, ginger, lemon
**Nutritional information (per 100 mL)**				
Energy (kCal)	0.5	36	0.25	7
Energy (kJ)	1.9	151	2	29
Fat (g)	0	0	0	< 0.1
Protein (g)	0.05	0	0	< 0.1
Carbohydrates (g)	0.1	8.6	0	1.5
including sugars	0	8.6	0	0
Sodium (mg)	1.0	6.7	12	< 5

* Components reported as per manufacturers’ nutritional information labels.

**Table 2 nutrients-14-02758-t002:** Female group means ± SEM for sucrose exposure in Phase 1 and NNS exposure in Phase 2.

*Phase 1 Sucrose Exposure*	*Control* (*n* = 8)	*Diet Coke* (*n* = 8)	*Sprite* (*n* = 8)	*Cordial* (*n* = 8)	*Kombucha* (*n* = 8)
Baseline BW (g)	142.6 ± 4.4	141.9 ± 4.1	141.7 ± 4.1	142.1.1 ± 3.7	142.4 ± 3.6
Chow intake (g/rat/day)	11.4 ± 0.4	11.5 ± 0.3	11.3 ± 0.4	12.8 ± 0.5	12.2 ± 0.4
Sucrose (mL/rat/day)	78.6 ± 1.4	81.5 ± 1.3	89.9 ± 3.0	75.6 ± 4.0	72.4 ± 2.4
Water (mL/rat/day)	0.7 ± 0.2	0.5 ± 0.1	0.7 ± 0.01	0.7 ± 0.01	0.7 ± 0.01
Total energy intake (kcal/rat/day)	70.3 ± 1.3	70.6 ± 1.1	74.6 ± 0.9	73.6 ± 1.1	70.5 ± 1.2
Fasting blood glucose (mM)	4.7 ± 0.1	4.7 ± 0.1	4.8 ± 0.1	5.0 ± 0.1	4.7 ± 0.1
** *Phase 2 NNS exposure* **					
Chow intake (g/rat/day)	14.0 ± 0.7 ^b^	14.7 ± 0.2	11.4 ± 0.4 *	15.3 ± 0.4	15.8 ± 0.5 ^a^
Test drink (mL/rat/day)	N/A	17.5 ± 1.5	50.9 ± 1.6	6.8 ± 0.6	9.6 ± 0.4
Water (mL/rat/day)	21.4 ± 0.4	10.5 ± 0.7	2.6 ± 0.3	18.8 ± 0.6	20.3 ± 0.8
Total fluid intake (mL/rat/day)	21.4 ± 0.4	28.0 ± 0.9	53.5 ± 1.7	25.7 ± 0.4	30.0 ± 0.9
Total energy intake (kcal/rat/day)	47.3 ± 0.9	49.9 ± 0.7	56.7 ± 0.7	51.3 ± 0.7	53.8 ± 0.8
Fasting blood glucose (mM)	4.6 ± 0.1	4.8 ± 0.1	5.1 ± 0.1 *	4.7 ± 0.1	4.7 ± 0.1
Fasting plasma insulin (ng/mL)	NR	NR	NR	NR	NR
Retroperitoneal fat mass (g)	13.0 ± 0.9	17.6 ± 2.2	16.0 ± 1.7	17.8 ± 2.5	18.2 ± 2.3

Abbreviations: SEM—standard error of mean; n—sample size; BW—body weight; g—grams; mL—millilitres; kcal—kilocalorie; mM—millimolar; NNS—non-nutritive sweetener; N/A—not applicable; ng—nanogram; NR—not reportable. * Significantly different from all the other groups, *p* < 0.05. ^a^ Significantly different from ^b^. Note: The control group in Phase 1 was provided sucrose and water, in Phase 2—water only.

**Table 3 nutrients-14-02758-t003:** Male group means ± SEM for sucrose exposure in Phase 1 and NNS exposure in Phase 2.

*Phase 1 Sucrose Exposure*	*Control* (*n* = 8)	*Diet Coke* (*n* = 8)	*Sprite* (*n* = 8)	*Cordial* (*n* = 8)	*Kombucha* (*n* = 8)
Baseline BW (g)	247.6 ± 5.9	243.9 ± 5.0	248.4 ± 3.4	248.1 ± 5.2	244.6 ± 4.6
Chow intake (g/rat/day)	22.5 ± 0.6	22.0 ± 0.5	20.6 ± 0.7	21.9 ± 0.6	21.9 ± 0.5
Sucrose (mL/rat/day)	101.7 ± 4.7	85.2 ± 1.5	90.1 ± 4.1	77.0 ± 1.6	119.9 ± 7.4
Water (mL/rat/day)	0.8 ± 0.1	0.8 ± 0.03	1.0 ± 0.1	1.0 ± 0.2	1.1 ± 0.2
Total energy intake (kcal/rat/day)	115.9 ± 1.4	108.1 ± 1.8	105.2 ± 2.0	104.6 ± 2.3	122.6 ± 1.9
Fasting blood glucose (mM)	4.8 ± 0.1	5.1 ± 0.1	4.9 ± 0.1	5.2 ± 0.2	5.2 ± 0.1
Fasting plasma insulin (ng/mL)	0.7 ± 0.1	1.0 ± 0.1	1.0 ± 0.1	1.1 ± 0.1	1.0 ± 0.1
* **Phase 2 NNS exposure** *					
Chow intake (g/rat/d)	25.9 ± 0.2	24.6 ± 0.2	20.9 ± 0.2 *	24.7 ± 0.3	24.8 ± 0.2
Test drink (mL/rat/day)	N/A	26.0 ± 1.8	77.5 ± 1.8	9.0 ± 0.7	12.6 ± 1.2
Water (mL/rat/day)	33.5 ± 0.4	13.5 ± 0.7	3.6 ± 0.3	24.1 ± 0.6	31.9 ± 0.6
Total fluid intake (mL/rat/day)	33.5 ± 0.4	39.5 ± 1.3	81.1 ± 2.0	33.1 ± 0.4	44.5 ± 1.2
Total energy intake (kcal/rat/day)	88.1 ± 0.8	83.7 ± 0.7	99.0 ± 1.0 *	84.0 ± 1.0	85.1 ± 0.9
Fasting BGL (mM)	5.5 ± 0.1	5.3 ± 0.1	5.2 ± 0.1	5.2 ± 0.1	5.2 ± 0.2
Fasting plasma insulin (ng/mL)	0.5 ± 0.1 ^b^	0.8 ± 0.1	1.0 ± 0.1 ^a^	0.9 ± 0.1	0.8 ± 0.1
Retroperitoneal fat mass (g)	25.1 ± 1.8	24.1 ± 1.8	23.9 ± 3.1	19.3 ± 1.5	21.4 ± 2.9

Abbreviations: SEM—standard error of mean; n—sample size; BW—body weight; g—grams; mL—millilitres; kcal—kilocalorie; mM—millimolar; NNS—non-nutritive sweetener; N/A—not applicable; ng—nanogram. * Significantly different from all the other groups, *p* < 0.05. ^a^ Significantly different from ^b^. Note: The control group in Phase 1 was provided sucrose and water, in Phase 2—water only.

## Data Availability

The data presented in this study are openly available in FigShare at 10.6084/m9.figshare.19920695, accessed on 3 June 2022.

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
