# Peer review of "An Ecological Validity Model for the Prevention of Obesity: Non-Nutritive Sweetener Consumption in Rats and the Effects of Switching from Sugar-Sweetened to Diet Beverages"

_nutrients, 2022, doi:10.3390/nu14132758_

Round 1
Reviewer 1 Report
This study presents an "An ecological validity model for the prevention of obesity: non-nutritive sweetener consumption in rats and the effects of switching from sugar sweetened to diet beverages". My comments are listed below:
Introduction
-Authors need to define obesity/overweight for humans (males vs females).
- Authors need to include additional data on global prevalence and impact of obesity in terms of premature deaths. The following site provides good information: https://ourworldindata.org/obesity#
Methods
- Did authors define what would constitute as "obesity" in male and female rats?
Discussion
- Authors need to include a paragraph on how their study results are relevant in context of the human obesity and what should be some future research directions.
Author Response
Point 1: Authors need to define obesity/overweight for humans (males vs females). (Introduction)
Response 1: We appreciate the reviewer suggesting the improvements to define obesity/overweight. This has now been included (Introduction; Page 1; Lines 40-43)
Point 2: Authors need to include additional data on global prevalence and impact of obesity in terms of premature deaths. The following site provides good information: https://ourworldindata.org/obesity# (Introduction)
Response 2: Additional data on prevalence and impact of obesity has been included. We thank the reviewer for providing an excellent site for data (Introduction; Page 1; Lines 36-38).
Point 3: Did authors define what would constitute as "obesity" in male and female rats? (Methods)
Response 3: In response to the reviewer’s question on the definition of ‘obesity’ in rats, we provide further clarification and amendments.
Currently in pre-clinical research, there is limited information on anthropometrical parameters and there is no standard definition of “obesity” or “adiposity” in laboratory rats (Novelli et. Al, 2007). This may be due to the number of differing tools used to measure adiposity, including but not limited body weight, body weight gain, varying fat pad masses and DEXA body scanning. Moreover, differing strains of rats may show differing effects from a dietary intervention, thus consistency in defining obesity in ‘rats’ may not be useful.
This has been further clarified in the methods section (Methods; Page 6; Lines 227-229).
Point 4: Authors need to include a paragraph on how their study results are relevant in context of the human obesity and what should be some future research directions. (Discussion)
Response 4: The reviewer raises an important point given one of aims of this study was to enhance translation, therefore it is crucial to discuss the relevance of the animal results in the context of human obesity. We have included a paragraph in the discussion section outlining how we made best efforts to enhance translation by addressing external validity. Therefore our results may be relevant when applying strategies in humans, including recommendations of replacing sugar sweetened beverages with diet beverages. Further, this paragraph includes suggested directions for future research (Discussion; Page 15 and 16; Lines 531-554).
Reviewer 2 Report
This study aims to determine if rats consume commercially available NNS beverages so an ecologically valid model for NNS consumption could be developed, and to determine if commercially available beverages sweetened with NNSs impact metabolic recovery from the damage caused by excess consumption of sugar.
The authors concluded that an ecologically valid model for NNS consumption can be developed for some commercially available NNS beverages and that no group difference was observed in retroperitoneal fat mass, body weight change or FBGL, suggesting both sexes exhibited limited metabolic recovery.
This article is well written and of scientific interest.
I have some concerns.
Major comments
1 Although the authors concluded that no group difference was observed in retroperitoneal fat mass, body weight change or FBGL, suggesting both sexes exhibited limited metabolic recovery, does it indicate that NNSs do not aggravate glucose metabolism and obesity? If so, it may be better to describe such interpretations.
2 It is unclear whether the degrees of sweet taste were adjusted between NNSs and sucrose groups.
If adjusted, please describe how to adjust the degrees of sweet taste between groups in the methods section.
3 It may be natural that Sprite was more favorite for the rats than other NNSs because it contains sucrose, which is more fullness.
4 I wonder that the NNSs except Sprite were nonacceptable for the rat compared with sucrose, which is indicated by Figure 4. Sprite includes sucrose, thereby it leads to acceptable for the rat.
Please discuss this issue.
5 In Figure 4, is the data of sucrose group in the phase 2 (continuous group) available? It may be interesting to compare switch and continuous sucrose.
Minor comments
1 In the table 2 and 3, it is unclear that control group means water group or sucrose group. Just in case, please describe it clearly.
Author Response
Major comments
Point 1: Although the authors concluded that no group difference was observed in retroperitoneal fat mass, body weight change or FBGL, suggesting both sexes exhibited limited metabolic recovery, does it indicate that NNSs do not aggravate glucose metabolism and obesity? If so, it may be better to describe such interpretations.
Response 1: We thank the reviewer for raising this important point. We have pondered heavily on this possibility and agree that our study cannot rule out the fact the limited metabolic recovery may in fact be impacted by NNS; albeit only small volumes of kombucha and cordial were consumed. We have incorporated this point into the discussion (Discussion; Page 15; Lines 515-521).
Point 2: It is unclear whether the degrees of sweet taste were adjusted between NNSs and sucrose groups.
If adjusted, please describe how to adjust the degrees of sweet taste between groups in the methods section.
Response 2: In response to the reviewer’s question, we did not adjust sweet taste between the groups. As our primary aim was to determine if rats drink commercially available beverages, the preference and acceptance tests performed prior to and post sucrose feeding were done in an identical manner for each group.
Point 3: It may be natural that Sprite was more favorite for the rats than other NNSs because it contains sucrose, which is more fullness.
Response 3: This has been included in line with the reviewer’s comment (Discussion; Page 14; Lines 465-466).
Point 4: I wonder that the NNSs except Sprite were nonacceptable for the rat compared with sucrose, which is indicated by Figure 4. Sprite includes sucrose, thereby it leads to acceptable for the rat.
Please discuss this issue.
Response 4: We thank the reviewer for raising this point. For further clarification, acceptance testing in rodents is standardly defined as the volume of intake over a short period (Duca, Swartz & Cuvasa, 2014). In this instance, we tested our rats over a 4-hr period. If a rat consumes any volume over 1mL (which we defined as the error for spillage), this would be determined as “accepting” the beverage. This was described in Methods section (Page 5; Lines 188-189).
Rodents have been shown to ‘accept’ various non-caloric NNS solutions in many studies, including saccharin, stevia and aspartame. Our data identified that all NNS beverages were accepted, albeit at differing levels (i.e. diet coke was more accepted than kombucha and cordial). We accept the premise that Sprite was more acceptable than the other groups as the volumes were significantly greater than other groups, however according to the definition, we reject the statement that the other NNS groups were nonacceptable.
In regard to preference testing, that is which beverage is preferred over another, our data showed sprite was highly preferred to water, whereas for the other groups, water was preferred. This does not exclude the fact that rats will still accept the fluid.
Point 5: In Figure 4, is the data of sucrose group in the phase 2 (continuous group) available? It may be interesting to compare switch and continuous sucrose.
Response 5: Due to constraints in line with the “Reduction” in the 3R’s of animal research, along with laboratory housing and financial considerations, we did not include a continuous sucrose group; therefore, data is not available. We agree it would be an interesting comparison and have suggested this in future research considerations (Discussion; Page 16; Lines 553-555).
Minor comments
Point 6: In the table 2 and 3, it is unclear that control group means water group or sucrose group. Just in case, please describe it clearly.
Response 6: This has been now been clearly described in the footnote of both Table 2 and Table 3 (Results; Page 11; Lines 354-355 and 362-363).
Round 2
Reviewer 2 Report
The manuscript has been improved.